# *RT-DOb*, a switch gene for the gene pair {*Csf1r, Milr1*}, can influence the onset of Alzheimer's disease by regulating communication between mast cell and microglia

**Nasibeh Khayer**[1]*, **Nasrin Motamed**[2], **Sayed-Amir Marashi**[3]*, **Fatemeh Goshadrou**[4]*

**1** Skull Base Research Center, The Five Senses Health Institute, School of Medicine, Iran University of Medical Sciences, Tehran, Iran, **2** Department of Cellular and Molecular Biology, School of Biology, University of Tehran, Tehran, Iran, **3** Department of Biotechnology, College of Science, University of Tehran, Tehran, Iran, **4** Department of Basic Sciences, Faculty of Paramedical Sciences, Shahid Beheshti University of Medical Sciences, Tehran, Iran

* fgoshadrou@sbmu.ac.ir (FG); marashi@ut.ac.ir (SAM)

**Data Availability Statement:** All relevant data are within the paper and its Supporting Information files.

## Abstract

In biology, homeostasis is a central cellular phenomenon that plays a crucial role in survival. The central nervous system (CNS) is controlled by exquisitely sensitive homeostatic mechanisms when facing inflammatory or pathological insults. Mast cells and microglia play a crucial role in CNS homeostasis by eliminating damaged or unnecessary neurons and synapses. Therefore, decoding molecular circuits that regulate CNS homeostasis may lead to more effective therapeutic strategies that specifically target particular subsets for better therapy of Alzheimer's disease (AD). Based on a computational analysis of a microarray dataset related to AD, the *H2-Ob* gene was previously identified as a potential modulator of the homeostatic balance between mast cells and microglia. Specifically, it plays such a role in the presence of a three-way gene interaction in which the *H2-Ob* gene acts as a switch in the co-expression relationship of two genes, *Csf1r* and *Milr1*. Therefore, the importance of the *H2-Ob* gene as a potential therapeutic target for AD has led us to experimentally validate this relationship using the quantitative real-time PCR technique. In the experimental investigation, we confirmed that a change in the expression levels of the *RT1-DOb* gene (the rat ortholog of murine *H2-Ob*) can switch the co-expression relationship between *Csf1r* and *Milr1*. Furthermore, since the *RT1-DOb* gene is up-regulated in AD, the mentioned triplets might be related to triggering AD.

## Introduction

Homeostasis is a pivotal cellular phenomenon with crucial roles in survival [1]. Accordingly, the central nervous system (CNS) is controlled by exquisitely sensitive homeostatic mechanisms when faced with inflammatory or pathological insults. Mast cells and microglia serve a crucial role in CNS homeostasis by purging damaged or unnecessary neurons and synapses.

**Funding:** The author(s) received no specific funding for this work.

**Competing interests:** The authors have declared that no competing interests exist.

**Abbreviations:** CNS, Central nervous system; AD, Alzheimer's diseases; MVM, Morris water maze; SEM, Standard error of the mean; H2-Ob, Histocompatibility 2, O region beta locus; Slc14a1, Solute carrier family 14 member 1; Milr1, Mast cell immunoglobulin like receptor 1; Csf1r, Colony stimulating factor 1 receptor; Hexb, Hexosaminidase subunit beta; Slamf6, Signaling lymphocytic activation molecule family member 6.

Mast cells are the first responder cells to brain injury and they have a critical role in beginning the inflammation of the CNS [2]. Inflammation mediators such as histamine [3] and tryptase [4] that are released by mast cells can induce microglial activation. Activated mast cells can also directly cause the activation of microglia [5]. Activated microglia release neurotrophin [6]. In response to exogenously induced stress such as inflammation, palmitoylethanolamide, a microglia-specific molecule [7], can promote cellular homeostasis by inhibiting mast cell activity [8, 9]. Additionally, microglia possess endogenous homeostatic molecules/ mechanisms that can be up-regulated in response to tissue destruction or induction of inflammatory responses.

Previously, putative switch mechanisms involved in triggering Alzheimer's disease (AD) were detected using a three-way interaction model [10]. Use of three-way interaction models, including the liquid association model that was used in the earlier study, is an efficient approach for finding switch mechanisms [10–14]. In this model, it is assumed that there is a switch gene, say $Z$, which controls the co-expression of a gene pair, say {$X_1$ and $X_2$}. This triplet, from now on, will be shown as $Z$/{$X_1$ and $X_2$}.

Previously, we identified two statistically significant triplets based on the computational analysis of a murine AD-related gene expression microarray dataset [15]. It was suggested that *H2-Ob* may act as the switch gene for the {*Csf1r*, *Milr1*} gene pair, while *Slc14a1* may act as the switch gene for the {*Slamf6*, *Hexb*} gene pair. Based on the results of previous studies, it was suggested that the homeostasis-related link between mast cells and microglia may be controlled by the expression level of the *H2-Ob* gene. Our analysis, however, did not specifically reveal the AD-related biological relevance of the *Slc14a1*/{*Slamf6*, *Hexb*} triplet.

## On the role of *H2-Ob*, *Csf1r*, *Milr1*, *Slc14a1*, *Slamf6* and *Hexb* in AD

The *H2-Ob* gene, also known as H2-O in humans, encodes a non-classical major histocompatibility complex (MHC) class II molecule that is expressed specifically by antigen-presenting cells such as macrophages, dendritic cells, and B cells [16]. Some evidence suggests that the *H2-Ob* gene may play a role in neurodegenerative diseases, including AD [17, 18]. It has been found that *H2-Ob* is up-regulated in Niemann-Pick disease type C and multiple sclerosis, two neurodegenerative disorders [19, 20]. Moreover, a recent study showed that the expression levels of MHC class II molecules were increased in a transgenic rat model of human tauopathy [21]. Furthermore, using a computational model, we previously suggested that the *H2-Ob* gene might regulate the co-expression relationship of two genes *Csf1r* and *Milr1*, possibly being involved in AD [15]. In addition, some experimental studies suggest that HLA-DOB is involved in AD [22]. These findings suggest that the *H2-Ob* gene may play a critical role in the pathogenesis of AD through its involvement in the immune response and microglial function. In addition, modulating the activation of non-neuronal cells that normally control neuronal sensitization, for example by targeting *H2-Ob* gene expression, opens new perspectives for the development of therapies that target neuroinflammation and AD progression [23]. Therefore, further research is needed to fully understand the mechanisms underlying the role of *H2-Ob* gene in the course of neurodegeneration.

The *Csf1r* gene, also known as colony-stimulating factor 1 receptor gene, plays a well-established role in the differentiation and survival of mononuclear phagocytes, including microglia in the CNS [24]. Several studies have shown that dysregulation of the *Csf1r* gene and its associated signaling pathway is linked to microglial dysfunction and, in turn, the pathogenesis of neurodegenerative diseases. More specifically, pharmacological inhibition of the *Csf1r* gene reduces overall neuroinflammation and prevents neuronal loss and memory impairment in the AD mice model [25, 26]. Another study showed that genetic variants of the *Csf1r* gene

were associated with an increased risk of AD [27]. In addition, inhibition of CSF-1R has been proposed as a potential therapeutic strategy for neurodegenerative diseases, including AD [28–30].

The Mast Cell Immunoglobulin-Like Receptor 1 gene (MILR1) encodes the Allergin-1 protein, which is expressed on the surface of mast cells [31]. Allergin-1 contains an immunoreceptor tyrosine-based inhibitory motif (ITIM)-like domain that modulates mast cell activation via the FcεRI-mediated signaling pathway [32]. Although Allergin-1 is not reported to plays a role in the pathogenesis of neurodegenerative diseases, existing evidence suggests its potential indirect involvement. Amyloid peptides, as a hallmark of AD pathology, induce mast cell activation and degranulation response, resulting in the release of different pro-inflammatory substances [33, 34]. The release of such bioactive molecules may be associated with the onset (or even progression) of AD [35, 36]. On the other hand, Allergin-1 acts as a modulator protein for mast cell activation and also inhibits mast cell degranulation via suppression of FcεRI-mediated signaling [32]. Furthermore, the dysregulation of the FcεRI-mediated signaling pathway in AD pathology has been reported recently [15, 37]. Taken together, the role of the *Milr1* gene (Allergin-1) in neurodegenerative diseases, especially AD, is sensible.

The *Slamf6* gene, also known as CD352, is a member of the signaling lymphocyte activation molecule (SLAM) family of receptors. It is predominantly expressed on the surface of natural killer (NK) cells, T cells, and dendritic cells. The Slamf6 gene is involved in various physiological and pathological processes, regulating immune responses, NK-cell development and cytotoxicity, neutrophil functions, and trogocytosis [38]. Although the Slamf6 gene has been identified as a contributing factor to the progression of some autoimmune diseases [39–41], limited research has been done on its direct involvement in neurodegenerative diseases. However, a recent study suggested that dysregulation of in the Slamf6 gene may contribute to the development of multiple sclerosis [42].

The *Hexb* gene encodes the beta subunit of β-hexosaminidase, which is stably expressed in brain microglia. This lysosomal enzyme is involved in the breakdown of gangliosides. Mutations in the *Hexb* gene can result in the accumulation of glycosphingolipids, leading to various lysosomal storage disorders collectively known as GM2 gangliosidosis [43]. Lysosomal abnormalities are one of the hallmarks of AD, and they progress over time. According to a study, the accumulation of ganglioside-bound amyloid peptide, which is associated with AD, occurs due to lysosomal dysfunction in the β-hexosaminidase knockout mouse model [44]. Another study found that although heterozygosity of the Hexb gene reduced learning flexibility, such gene is haploinsufficient in the mouse AD brain [45]. Recent studies reveal that the *Hexb* gene is up-regulated in the AD brain at both transcriptome [46] and proteome levels [47].

The *Slc14a1* gene, also known as the urea transporter UT-B, is responsible for the transport of urea across cell membranes in various organs, including the brain. The SLC14A1 gene is primarily expressed in astrocytes and ependymal cells, in which it is believed to play a central role in the maintenance of urea homeostasis of the nervous system [48]. Although the precise role of the *Slc14a1* gene in neurodegenerative diseases remains poorly understood, several studies have reported the dysregulation of Slc14a1 expression in neurodegenerative diseases, including AD [15], HD [49] and ALS [50]. Moreover, disruption of urea metabolism and subsequent accumulation of urea in the brain has been reported in AD [51], whilst the Aβ-derived ammonia is reported to be detoxified by the urea cycle of astrocytes [52]. Furthermore, *Slc14a1*-deficient mice exhibit urea accumulation in the hippocampus, resulting in behavioral impairment, nitric oxide disruption and neuronal loss [53]. A recent study indicated that the expression level of the *Slc14a1* gene regulates the inflammatory responses in microglial and neuroblastoma cells, following lipopolysaccharide exposure. Therefore, it has been suggested that the inhibition of the *Slc14a1* gene can be a potential therapeutic target for AD [54].

### Rodent models of AD

Amyloid-β (Aβ) is the primary pathological hallmark of familial and sporadic AD. The accumulation of Aβ oligomers in the brain can lead to the formation of plaques, which contribute to neuronal damage and cognitive decline. Clinical and experimental studies provide strong evidence that the acute increase in Aβ levels in the brain is responsible for inducing AD-like phenotypes [55, 56]. Therefore, animal models of AD have been developed based on neural overexpression or administration of Aβ. Two commonly used animal models to study AD are transgenic and non-transgenic rodent models. While transgenic rodent models are useful in mimicking human AD pathology, their development is time-consuming and expensive. Non-transgenic models, such as pathogen-induced AD models, can produce AD-like behavioral abnormalities, although may not accurately reflect the fundamental pathophysiology of AD. The Aβ-injected rodent model is a pathogen-induced AD model that exhibits Aβ pathology and is attractive for AD investigation due to its controllability. Researchers can control individual differences in the mice and enable timely drug treatment depending on the mechanism of the candidate drug [57]. Moreover, injecting Aβ peptide into the hippocampus has been associated with several characteristic features of AD, including inflammatory reactivity, neuronal loss, and vascular perturbations [58, 59]. Specifically, intra-hippocampal injection of Aβ peptide leads to inflammatory responses mediated by activated microglia, which is coupled with granule cell neuron loss in the rat AD model [60]. Finally, previous studies suggest that intracerebral injection of Aβ peptide can change the gene expression profile [61].

In the present study, we aimed to assess, by empirical studies, the validity of the computationally-discovered relation between the genes of both *H2-Ob*/{*Csf1r*, *Milr1*} triplet and the *Slc14a1*{*Slamf6*, *Hexb*} triplet. Furthermore, we used a rat Aβ model of AD, a non-transgenic model, to detect expression levels of above genes.

## Materials and methods

### Animals

Adult male Wistar rats (200 ± 20 g) were used in this study. Rats used because of lack of access to mice strains that had been used in the microarray study [15]. The rats were housed in a temperature- and humidity- controlled vivarium under a 12-hour light/dark cycle, with free access to food and water. They were allowed to habituate in their cages for at least one week before the initiation of the experiment. All the experiments were conducted in accordance to the Guide for the Care and Use of Laboratory Animals [62]. The study was approved by the Research and Ethics Committee of Shahid Beheshti University of Medical Sciences (permission code: IR.SBMU.RAM.REC.1394.577).

### Creation of a rat Aβ model of Alzheimer's disease

Aβ1–42 peptide (Sigma Aldrich, St. Louis, Mo, USA) was dissolved in a 0.9% saline solution at a concentration of 5 mg/ml and subsequently incubated for one week at 37˚C to induce aggregate formation. Rats were anesthetized with an injection of ketamine (70 mg/kg) and xylazine (10 mg/kg) intraperitoneally. Then, by stereotaxic injection, 2 μl (10 μg) of Aβ1–42 oligomers was delivered into the lateral ventricle of each of the nine rats (AD model) over the course of approximately 5 minutes. The injection site coordinates (AP = -1.2 ± 0.2 mm; ML = 1.5 ± 0.5 mm; DV = -4 mm) was determined according to the rat brain atlas [63]. Co-application of trypan blue confirmed the injection site. Nine rats of the control group were subjected to the same procedure, except that they were injected with an isotonic saline solution.

## Histologic confirmation of Aβ plaque formation

The Congo red histology was carried out to confirm the formation of Aβ plaques [64]. Twenty days after injections into the brain, three randomly selected rats from each group were deeply anesthetized with a lethal dose of sodium pentobarbital. The brains of the animals were removed and immersed in 10% buffered formalin for 48–72 h. Then, dehydration and paraffin embedding were performed using an automated processor. The brains were then sectioned and stained with Congo red according to the standard protocol [65].

## The Morris water maze behavioral test

The Morris water maze (MWM) behavioral test [66] was used to assess spatial learning memory. The MVM test consists of two parts, namely, the "place navigation test" and the "spatial probe test". The place navigation test is used to evaluate learning by measuring the escape latency and searching distance to find the escape platform. The spatial probe test is used to evaluate the memory function for finding the removed escape platform. The place navigation test consists of 12 trials (4 trials/day for three days) and was followed by one probe test. The details of the tests have been previously described [67]. The first navigation test was performed twenty days after the brain injections.

## RNA extraction, cDNA synthesis, and RT-qPCR

Total RNA was extracted and collected from the hippocampus of each rat using the TriPure Isolation Reagent (Roche Applied Sciences) according to the manufacturer's instructions. For each sample, the purity of the RNA was confirmed by measuring optical density at 260 and 280 nm, and the integrity of the RNA was verified by the detection of 18S and 28S ribosomal RNA bands after gel electrophoresis. The RNA was used as template for cDNA synthesis with random primers using the RevertAid First Strand cDNA Synthesis Kit (Fermentas, Thermo Fisher Scientific, Korea) according to the manufacturer's instructions. The cDNAs were stored at -40˚C until use in RT-qPCR experiments.

RT-qPCR analysis was carried out to measure the levels of the mRNAs of six selected genes in the hippocampus of the AD and the control rats. The genes included *RT1-Dob*, *Csf1r*, *Milr1*, *Slc14a*, *Slamf6*, and *Hexb*. Oligonucleotide primers were designed based on GenBank reference sequences using the OLIGO 7 software [68] (Table 1). The 18S rRNA encoding gene served as the housekeeping gene in the analyses. RT-qPCR amplification reactions were performed using the Fast-Start DNA Master SYBR-Green I kit (Roche Applied Science, Germany) in a MIC (Magnetic Induction Cycler) machine (BioMolecular Systems, Australia). All reactions were carried out in triplicate.

## Statistical analysis

The data are expressed as mean± standard error of the mean (means ± SEM). The two-way repeated-measures analysis of variance (two-way ANOVA) was used to examine the effects of Aβ treatment and time (as the two independent variables) on two dependent variables separately, including the escape latency time and searching distance. The memory difference between the AD model and control groups was compared using the Mann-Whitney test (since the Kolmogorov-Smirnov test suggested that the distribution of the Morris water maze data were not normally distributed). The pairwise co-expression of surveyed genes was determined by Pearson's correlation coefficient ($r$). A $p$-value of less than 0.05 was considered statistically significant. All statistical analyses were performed in R.

**Table 1. Primer sequences used in RT-qPCR and real-time–PCR.**

| Gene | Reference Sequence ID | Forward/Reverse | Sequence | Amplicon Length |
|---|---|---|---|---|
| *Csf1r* | NM_001029901.1 | Forward | 5'– TGCACCGGAGAACATATACAG –3' | 139 |
| | | Reverse | 5'– CAGGTTAGCATAGTCCTGCTC –3' | |
| *Milr1* | NM_021585.1 | Forward | 5'– AACAAGAAGGCCTTGAGGATAG –3' | 136 |
| | | Reverse | 5'– TGAATTTCCTGTTTCTTCGTAAGC –3' | |
| *RT1-DOb* | NM_001008846.1 | Forward | 5'– CTTGGGCTGACTGTCTTTCT –3' | 96 |
| | | Reverse | 5'– CGCCCTTGAAGCCTCATT –3' | |
| *Hexb* | NM_001011946.1 | Forward | 5'– GCTCTGGAGCCCTAGAATTATC –3' | 89 |
| | | Reverse | 5'– CTATTCCGCGGCTGACC –3' | |
| *Slamf6* | NM_001191932.1 | Forward | 5'– TTCCTCTGACTCGCCAACAT –3' | 105 |
| | | Reverse | 5'– TCATTTCCTGCATCGGGTGA –3' | |
| *Slc14a1* | NM_019346.2 | Forward | 5'– AGATATCTTCTCTGGGCTCTGG –3' | 134 |
| | | Reverse | 5'– CTCCAAAGTAGGCAGTGAACAG –3' | |
| *Rn 18s* | NR_046237.1 | Forward | 5'– GGACACGGACAGGATTGACA –3' | 50 |
| | | Reverse | 5'– ACCCACGGAATCGAGAAAGA –3' | |

## Results

### Histologic confirmation of Aβ plaque formation in the hippocampus of AD model rats

Congo red staining of hippocampus tissue sections of AD model rats confirmed the presence of Aβ plaque formation (Fig 1). In contrast, Aβ plaques were not observed in the hippocampus of control rats. These results confirm that Aβ1–42 injection successfully induces the AD-like histological features in the rat brains.

**Confirmation of behavioral disruption in the AD model rats.** The searching distance in the place navigation test of both the AD model rats and the control rats decreased as the trial progressed (Fig 2A). However, the searching distance at all the time points was less in the control rats. Moreover, the AD model rats displayed a significant decrease in the probe quadrant's swimming distance compared to the control group in the probe test ($p$-value = 0.014) (Fig 2B). Similarly, the escape latency of both the AD model rats and the control rats decreased as the trial progressed, but the latency at all-time points was less in the control rats (Fig 2C). Additionally, the control rats needed less time to locate the hidden platform than the AD model

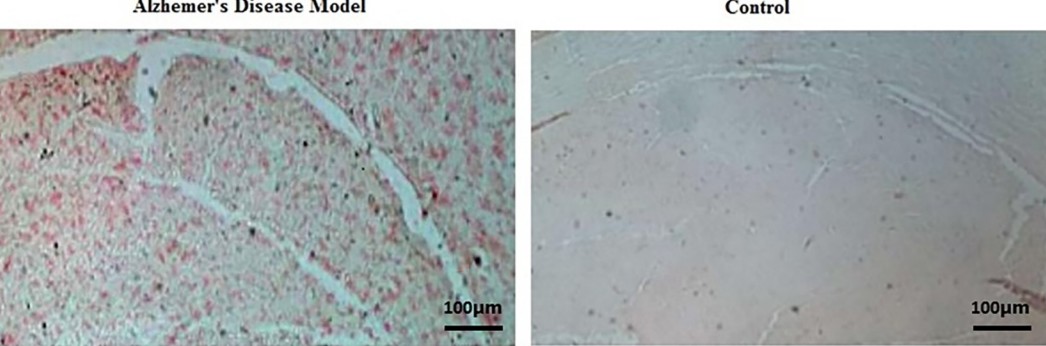

**Fig 1. Histological confirmation of Aβ plaque formation in the hippocampus of Alzheimer's disease model rats by Congo red staining.** The scale bar is 100 μm.

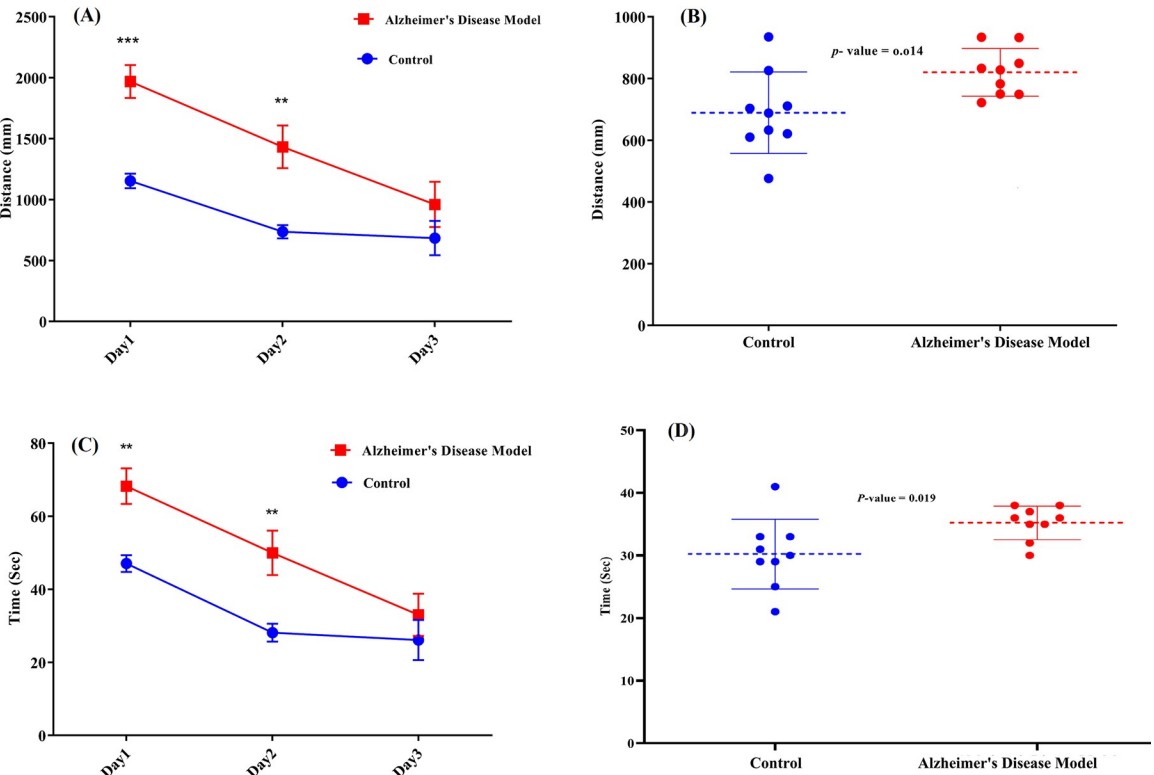

**Fig 2. Comparison of results of the Morris water maze test of the AD model rats and the control rats.** (A) Comparison of the searching distance in the place navigation test; (B) Comparison of the swimming distance in the target quadrant in the probe test; (C) Comparisons of the escape latency (*i.e.*, swimming time) in the place navigation test; (D) Comparison of the time spent in the target quadrant in the probe test. Data on each day are expressed as mean ± standard error.

rats in the probe test (*p*-value = 0.019) (Fig 2D). In total, the MVM behavior test results are consistent with decreased learning ability and decreased memory in the AD model animals.

## Assessment of relative expression levels of the genes of the *RT1-Dob*, *Csf1r*, and *Milr1* triplet and the genes of the *Slc14a1*, *Slamf6* and *Hexb* triplet

The observation of 18S and 28S rRNA bands upon electrophoresis of RNAs isolated from the hippocampus of the rats confirmed the integrity of extracted RNA (S1 File in S1 Appendix). Observation of single peaks following melting curve analysis of RT-qPCR products, and detection of single amplicon bands of expected size upon electrophoresis of amplification products of reverse transcripts of the RNAs were consistent with the specificity of the PCR primers (S2 File in S1 Appendix).

The fluorescent intensity of the cycle threshold (*Ct*) during RT-qPCR was considered as a measure of gene expression. The mean *Ct* values of replicate experiments for each gene are determined (presented in S3 File in S1 Appendix). The expression levels of the genes were normalized by comparison to the expression level of the housekeeping gene. The relationships between the normalized expression levels of the genes of each triplet were assessed. Additionally, the normalized expression levels of the putative switch genes in the AD model samples and the control samples were compared.

We found that when the normalized expression level of the *RT1-DOb* gene (that is, the rat counterpart of the *H2-Ob* gene) was in the range between 1.63 and 0.33, a direct correlation

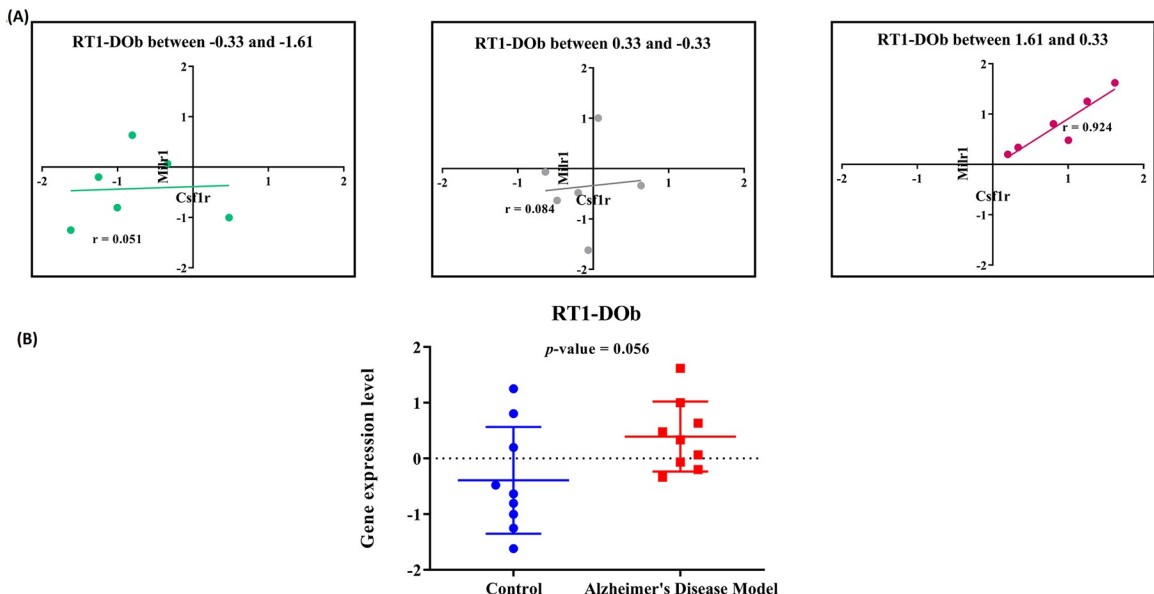

**Fig 3. Empirical assessment of the role of *RT1-DOb* as a switch gene for the {*Csf1r, Milr1*} gene pair in AD development.** (A) Comparison of normalized expression levels of *Csf1r* and *Milr1*A at different expression levels of *RT1-DOb*; (B) Comparison of expression levels of the putative switch gene *RT1-DOb* gene in the hippocampus of AD model rats and control rats.

between *Milr1* and *Csf1r* expression levels was observed ($r = 0.92$) (Fig 3A). In contrast, when the expression level of *RT1-DOb* was in the range between -1.63 and -0.33, the expression levels of the {*Milr1*, *Csf1r*} gene pair were not correlated ($r = 0.05$). In other words, the strength of correlation between the expression of *Milr1* and *Csf1r* was apparently affected by a change in expression level of *RT1-DOb* ($|r_{diff}| = 0.87$). These results are consistent with a potential switch role of the *RT1-DOb* for the {*Csf1r, Milr1*} gene pair. Interestingly, the expression level of the *RT1-DOb* gene in the AD hippocampus samples was higher than in control samples, and the difference approached statistical significance ($p = 0.056$) (Fig 3B). Overall, the findings are consistent with the proposition that *RT1-DOb* might act as a switch gene in promoting AD.

It should be noted here that although the change in expression level of the Z gene in a Z/{$X_1$, $X_2$} triplet is a fundamental characteristic of such a switch gene, it may not be directly associated with the effectiveness of the switch gene. In other words, sometimes, a slight change in the expression level of a switch gene (Z) can lead to a drastic shift in the co-expression of the $X_1$ and $X_2$ gene pair.

With respect to the putative switch gene *Slc14a1*, when its normalized expression level was in the range between 1.63 and 0.33, a direct correlation between the normalized expressions of *Slamf6* and *Hexb* was observed ($r = 0.75$) (Fig 4A). On the other hand, when the expression level of *Slc14a1* was in the range between -1.63 and -0.33, an inverse correlation between the expressions of *Slamf6* and *Hexb* ws detected ($r = -0.27$). In other words, the strength and direction of the correlation between the expression of the two genes *Slamf6* and *Hexb* were affected by changes in expression levels of *Slc14a1* ($|r_{diff}| = 1.02$). This finding is consistent with the assumption of the switch role of *Slc14a1* for the co-expression of {*Slamf6, Hexb*} gene pair. However, and notably, the subtle difference between the expression level of *Slc14a1* in the hippocampus of the AD model rats and the control rats is not significantly significant ($p = 0.614$) (Fig 4B). Therefore, this observation does not prove the contribution of the *Slc14a1*/{*Slamf6, Hexb*} triplet to the onset of AD.

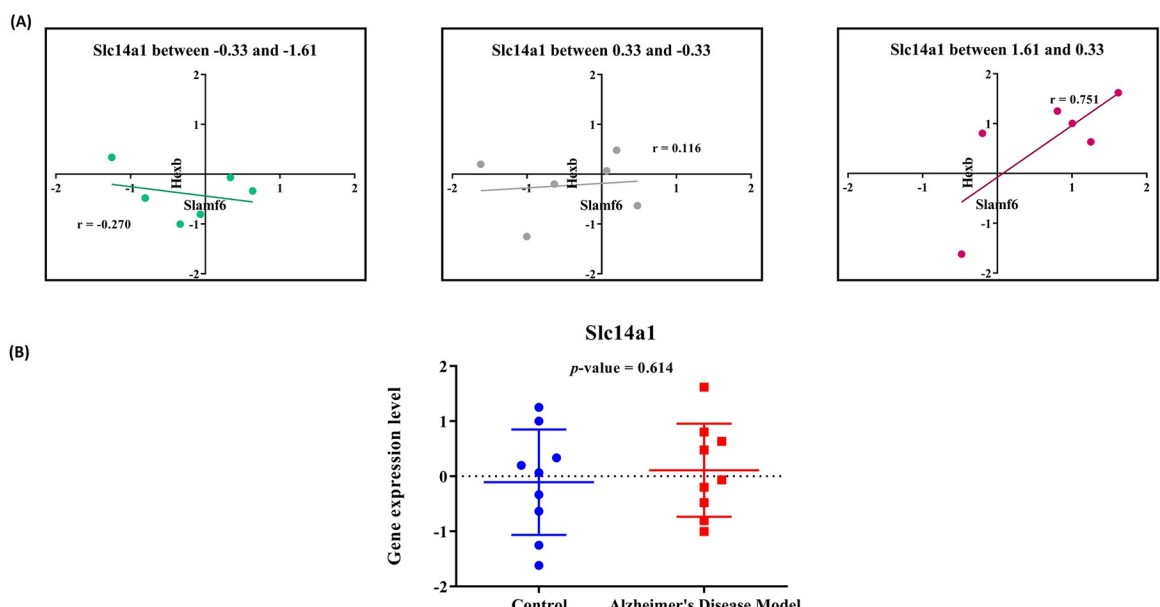

**Fig 4. Empirical assessment of the role of *Slc14a1* as a switch gene for the {*Slamf6, Hexb*} gene pair in AD development.** (A) Comparison of normalized expression levels of *Slamf6* and *Hexb* at different expression levels of *Slc14a1*; (B) Comparison of expression levels of the putative switch gene *Slc14a1*gene in the hippocampus of AD model rats and control rats.

## Discussion

In the present study, we experimentally evaluated the switch gene role of *RT1-DOb and Slc14a1* to control the co-expression of gene pairs {*Milr,Csf1r*} and {*Slamf6, Hexb*}, respectively. We also assessed the contribution of these two possible switch genes to the etiology of AD. These relationships and functions of the genes were surmised from an earlier computational study using the liquid association model [10]. The three-way interaction models, including the liquid association model, are more powerful tools to assess the interdependence of the gene expression levels. Our experimental findings are consistent for a potential role of *H2-Ob* both as a switch gene for *Milr* and *Csf1r*, and for potentially promoting AD.

We used a rat Aβ model of Alzheimer's disease. The Aβ peptide, the primary component of senile plaques in AD, contributes to neurotoxicity and other pathogenic mechanisms in various ways. Firstly, it can act as a transcription factor and directly regulate genes involved in AD-related pathways. Some pathways that can be directly regulated by Aβ peptide include insulin signaling, actin cytoskeleton dynamics, steroid and lipid metabolism, neuronal cell death, and DNA binding [69–72]. In particular, the Aβ peptide influences the expression of *Girk* and *Kcnq* channel genes, which are related to the impairment of learning and memory in AD [73].

Secondly, the Aβ peptide exerts its toxic effect indirectly by triggering intracellular cascades, resulting in the impairment of synaptic function, neuronal signaling, mast cell and microglial activities, energy metabolism, and mitochondrial function. For instance, the toxicity levels of Aβ peptide lead to tau phosphorylation, and in turn, neuronal death via the PAX6 signaling pathway [74]. Moreover, this peptide affects microglial activities via changes in the calcium level, proinflammatory activity, and gliotransmission through the RAGE-NF-κB pathway [75]. Furthermore, the Aβ oligomers are reported to induce memory impairment in an acute mouse model of AD through Toll-like receptor 4-dependent glial cell activation [76]. Additionally,

the Aβ peptide induces mast cell secretion via both a CD47/Pl-integrin membrane complex coupled with GJ-protein [34] and a pannexin1 hemichannel-dependent mechanism [33]. Also, the Aβ peptide activates downstream pathways related to mitochondrial damage in microglia through TREM2 (triggering receptor expressed in myeloid cells 2), which is coupled with aggravating inflammation and neurotoxicity [77].

Finally, there is strong evidence suggesting that Aβ peptide can play a pathogenic role in non-trivial ways. The Aβ peptide can independently induce uncontrolled, neurotoxic ion flux across cellular membranes, either by self-assembly to pores or by disrupting the lipid membrane. Specifically, exposure of the cell to Aβ leads to disrupting Ca2+ homeostasis and eventually elevated concentrations of intracellular calcium ions, which contribute to cognitive impairment and cell death [78, 79].

The results obtained from the Morris water maze (MWM) test clearly demonstrate a significant difference in the searching distance (Fig 2A) and the escape latency (Fig 2C) between the control and AD model groups on the first and second days of the training phase. As shown in the figure, the control group reached the escape platform in less time and covered a shorter searching distance compared to the AD model group. This difference was more pronounced on the first day, less so on the second day, and even less so on the third day. The difference on the third day was not statistically significant. These results indicate that training phase has an improving effect on learning in AD model rats. This effect has been confirmed in other studies as well [80]. Based on these findings, it is suggested that keeping the brain active through mental activities such as reading or solving puzzles may slow down the progression of Alzheimer's disease. However, the results of this study confirm the difference between the control and Alzheimer's groups not only on the first and second days but also on the spatial probe test (Fig 2B and 2D).

Studies that have directly investigated the role of the *H2-Ob* gene in microglia/mast cell response are rare, although some evidence suggests that it is important in microglial activation. The *H2-Ob* gene is a member of the major histocompatibility complex class II (MHC-II) antigens, which are expressed by antigen-presenting cells, including microglia [22]. MHC-II microglia plays a pivotal role in the activation of both innate and adaptive immune responses in neuroinflammation conditions, including AD [81–83]. Furthermore, dysregulation of *H2-Ob* may contribute to the pathogenesis of Parkinson's disease [84]. Although the above findings emphasize the role of MHC-II antigens in neuroinflammation pathogenesis, a conclusive association is yet to be found between *H2-Ob* expression and AD progression. Moreover, to the best of our knowledge, no experimental or computational evidence supports the claim that Aβ peptide regulates *H2-Ob* gene expression.

It was previously suggested that the homeostasis-related link between mast cells and microglia may be controlled by *H2-Ob* expression levels. *Milr1* and *Csf1r* are associated with the biological pathways "negative regulation of FcεRI-dependent signaling pathway" and "proliferation and activation of microglial cells", respectively [15]. Therefore, at those *H2-Ob* expression levels that *Milr1* and *Csf1r* are co-expressed, the above-mentioned biological processes may be activated in concert with each other. At the expression levels of *H2-Ob* where *Milr1* and *Csf1r* expression levels are inversely correlated, whoever, the two processes will not function together. Consequently, changes in the expression level of the *H2-Ob* gene can act as the switch factor to alter cellular behavior. Further investigations are needed to clarify the role of the *RT1-DOb* gene in the homeostasis-related link between mast cells and microglia.

With regards to *Slc14a1*, results of the empirical studies were consistent with its potential role as a switch gene for the {*Slamf6*, *Hexb*} gene pair. However, its role in AD etiology was not strongly supported by the experiments we performed in the rat model of the disease.

## Conclusion and future work

During inflammatory or pathological insults such as AD, The CNS is controlled by exquisitely sensitive homeostatic mechanisms. Therefore, decoding molecular circuits that regulate CNS homeostasis may lead to more effective therapeutic strategies that specifically target particular subsets for better therapy of AD. Previously, using a computational approach, it was suggested that *H2-Ob* expression levels in AD presumably control the homeostasis-related link between mast cells and microglia. Furthermore, it was suggested that the *H2-Ob* gene plays such a role via a statistically significant triplet (i.e., the *H2-Ob* gene and {*Csf1r*, *Milr1*} gene pair). The importance of the *H2-Ob* genes as a potential drug target for therapeutic AD led us to experimentally investigate the above relationship. Our results confirmed the switch role of the *RT1-DOb* (the rat orthologous of *H2-Ob*) gene in modulating the co-expression relationship of *Csf1r and Milr1* genes. Although the RT-qPCR technique verified the mentioned relationship, additional studies are needed to dissect the role of *H2-Ob* in mast cell-microglia communication. In the next step, the *H2-Ob* gene can be knocked down using a gene-knockdown technique such as RNA interference. Subsequently, the relationship between the *H2-Ob* gene and the {*Csf1r*, *Milr1*} gene pair be evaluated at the protein level using Western blotting. In addition, since the gene interactions involved in the two mentioned triplets have been defined in a mouse AD model [15], confirmation of those interactions in the human brain would be an importance step validating our findings. It should be noted here that switching mechanisms are associated with onset of disease. Therefore, the early stage of AD should be considered to validate these mechanisms.

## Supporting information

**S1 Appendix.**
(PDF)

## Acknowledgments

We would like to thank the University of Tehran, for providing us with research facilities.

## Author Contributions

**Conceptualization:** Nasibeh Khayer, Sayed-Amir Marashi, Fatemeh Goshadrou.

**Data curation:** Nasibeh Khayer, Nasrin Motamed.

**Investigation:** Nasrin Motamed, Fatemeh Goshadrou.

**Methodology:** Nasibeh Khayer, Nasrin Motamed, Fatemeh Goshadrou.

**Project administration:** Sayed-Amir Marashi.

**Supervision:** Nasrin Motamed, Sayed-Amir Marashi, Fatemeh Goshadrou.

**Validation:** Nasibeh Khayer, Fatemeh Goshadrou.

**Visualization:** Nasibeh Khayer.

**Writing – original draft:** Nasibeh Khayer.

**Writing – review & editing:** Sayed-Amir Marashi.

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
