## [Decision Letter · Decision Letter 0]

24 Jan 2023

PONE-D-22-24268RT-DOb  that is a switch gene for the gene pair {Csf1r, Milr1 } may affect Alzheimer’s disease onset by modulating mast cell-microglia communicationPLOS ONE

Dear Dr. Khayer,

Thank you for submitting your manuscript to PLOS ONE. After careful consideration, we feel that it has merit but does not fully meet PLOS ONE’s publication criteria as it currently stands. Therefore, we invite you to submit a revised version of the manuscript that addresses the points raised during the review process.

We look forward to receiving your revised manuscript.

Kind regards,

Chandler L. Walker

Academic Editor

PLOS ONE

Journal Requirements:

Additional Editor Comments:

The manuscript presented, "RT-DOb that is a switch gene for the gene pair {Csf1r, Milr1 } may affect Alzheimer’s disease onset by modulating mast cell-microglia communication" represents a meritorious study concerning the possible role of H2Ob as a genetic switch for the interaction between Csf1r and Milr1r genes in a model for Alzheimer's disease. Based on the comments from the reviewers, the manuscript will be re-evaluated after submission of a Major Revision.

Aspects of the following comments and suggests of particular importance are expanding on the background and rationale for using A-beta in the chosen rodent model, confirming and expanding details concerning the timeline of the A-beta expression data, and expansion of the Discussion section to better establish the genes switch mechanism and microglial/mast cell response. There needs to be more elaboration on the proposed mechanism of action in this process the limitations on the conclusions that can be made in using this rat model for this study.

Reviewer Comments:

The current work identifies a potential role for H2-Ob as a gene switch in the interaction between Csf1r and Milr1 gene pair. The work utilizes a rat model of AD, wherein stereotactic techniques were utilized to administer oligomerized A-beta into the hippocampus of rats. The sample sizes used (n=9 for both groups) were appropriate, and the authors utilize IHC, behavior and gene expression data to support their findings.

This review raises the following concerns about the manuscript and recommends the addressing of these concerns to the authors:

(1) Introduction: The introduction section is brief and would benefit from a literature review describing the five genes studied (namely, Csf1r, Milr1, H2-ob, Slamfb, Hexb).

(2) Introduction: Similarly, a stronger background should be provided on the utility of the A-beta administration model in rats; while the authors confirm A-beta presence in the hippocampus, it is unclear whether this results in microglial and mast cell activation in these rats.

(3) Results: The first three figures serve to establish the model (presence of a-beta and impact of a-beta on spatial learning), whereas the major claim of the paper are the subsequent figures. This reviewer recommends that the authors add additional figures associated with additional suggested work, and to add figures 1-3 (or some combination) to the supplementary data.

(4) Results: The data in Figure 4b is not statistically significant, but the conclusion seems to suggest that it is. This should be clarified.

(5) Results: While the authors clarify that they do not have access to AD mice models (hence the rationale for using the rat model), have the authors confirmed this gene interaction in post-mortem human brain? This would substantially strengthen their argument, as for now, it seems far too limited to a highly unique non-genetic murine model.

(6) Results: There is an incongruity in the timeline. The behavioral model in Fig 2 and 3 shows that the presumable inhibitory learning effects of A-beta are lost by day 3 – both tests show no different b/w the model and control at day 3. While the authors do not specify this (they should), it appears the gene expression data is performed at day 20 (after the brains were obtained for IHC). Therefore, there needs to be an explanation for the long-term effects of A-beta on the gene switch absent of any behavioral differences.

(7) Results: While the authors discuss the role of gene switch, there is a complete absence of confirmation of protein data. The gene switch role presupposes the ability of H2-ob to act as a transcription factor protein. Therefore, at the very least, protein expression via western blotting is necessary to complement the RT-PCR RNA expression data. If the proteins do not have antibodies available, perhaps an indirect measurement (measuring the activity/levels of a downstream pathway) would suffice as supporting evidence.

(8) Discussion: The discussion section is far too short and does not identify or speculate the mechanism of any of the work. The link between the gene switch and cellular response (microglia/mast) is minimal, and an identification of the method by which A-beta affects gene expression is entirely absent. For example, the section would benefit from an explanation of a-beta affecting cellular processes – either directly (there is data that supports the ability of a-beta to act as a transcription factor) or indirectly (via activation of intracellular cascades) or non-canonical explanations (such as data that supports the ability of a-beta to inset itself directly into the plasma membrane of neurons).

(9) Discussion: Is there computational or bioinformatics evidence that A-beta may regulate RT1-Dob gene expression data? Similarly, is there such data that RT1-Dob could mechanistically regulate the gene pair?

(10) Proof-read: This is a minor concern, but the authors might benefit from proofreading the manuscript. For example, the author email is not provided, and the end of the first intro paragraph has a stand alone “See below” at the end of it.

Overall, there is merit in the work performed by the authors. While the suggestions being made are non-trivial, it is anticipated that the authors will find the comments beneficial to their developing, valuable and interesting story about H2-Ob as a gene switch in the AD pathway.

Reviewers' comments:

Reviewer's Responses to Questions

**Comments to the Author**

1. Is the manuscript technically sound, and do the data support the conclusions?

Reviewer #1: Partly

2. Has the statistical analysis been performed appropriately and rigorously? 

Reviewer #1: Yes

3. Have the authors made all data underlying the findings in their manuscript fully available?

Reviewer #1: Yes

4. Is the manuscript presented in an intelligible fashion and written in standard English?

Reviewer #1: Yes

5. Review Comments to the Author

Reviewer #1: The current work identifies a potential role for H2-Ob as a gene switch in the interaction between Csf1r and Milr1 gene pair. The work utilizes a rat model of AD, wherein stereotactic techniques were utilized to administer oligomerized A-beta into the hippocampus of rats. The sample sizes used (n=9 for both groups) were appropriate, and the authors utilize IHC, behavior and gene expression data to support their findings.

This review raises the following concerns about the manuscript and recommends the addressing of these concerns to the authors:

(1) Introduction: The introduction section is brief and would benefit from a literature review describing the five genes studied (namely, Csf1r, Milr1, H2-ob, Slamfb, Hexb).

(2) Introduction: Similarly, a stronger background should be provided on the utility of the A-beta administration model in rats; while the authors confirm A-beta presence in the hippocampus, it is unclear whether this results in microglial and mast cell activation in these rats.

(3) Results: The first three figures serve to establish the model (presence of a-beta and impact of a-beta on spatial learning), whereas the major claim of the paper are the subsequent figures. This reviewer recommends that the authors add additional figures associated with additional suggested work, and to add figures 1-3 (or some combination) to the supplementary data.

(4) Results: The data in Figure 4b is not statistically significant, but the conclusion seems to suggest that it is. This should be clarified.

(5) Results: While the authors clarify that they do not have access to AD mice models (hence the rationale for using the rat model), have the authors confirmed this gene interaction in post-mortem human brain? This would substantially strengthen their argument, as for now, it seems far too limited to a highly unique non-genetic murine model.

(6) Results: There is an incongruity in the timeline. The behavioral model in Fig 2 and 3 shows that the presumable inhibitory learning effects of A-beta are lost by day 3 – both tests show no different b/w the model and control at day 3. While the authors do not specify this (they should), it appears the gene expression data is performed at day 20 (after the brains were obtained for IHC). Therefore, there needs to be an explanation for the long-term effects of A-beta on the gene switch absent of any behavioral differences.

(7) Results: While the authors discuss the role of gene switch, there is a complete absence of confirmation of protein data. The gene switch role presupposes the ability of H2-ob to act as a transcription factor protein. Therefore, at the very least, protein expression via western blotting is necessary to complement the RT-PCR RNA expression data. If the proteins do not have antibodies available, perhaps an indirect measurement (measuring the activity/levels of a downstream pathway) would suffice as supporting evidence.

(8) Discussion: The discussion section is far too short and does not identify or speculate the mechanism of any of the work. The link between the gene switch and cellular response (microglia/mast) is minimal, and an identification of the method by which A-beta affects gene expression is entirely absent. For example, the section would benefit from an explanation of a-beta affecting cellular processes – either directly (there is data that supports the ability of a-beta to act as a transcription factor) or indirectly (via activation of intracellular cascades) or noncanonical explanations (such as data that supports the ability of a-beta to inset itself directly into the plasma membrane of neurons).

(9) Discussion: Is there computational or bioinformatics evidence that A-beta may regulate RT1-Dob gene expression data? Similarly, is there such data that RT1-Dob could mechanistically regulate the gene pair?

(10) Proof-read: This is a minor concern, but the authors might benefit from proofreading the manuscript. For example, the author email is not provided, and the end of the first intro paragraph has a stand alone “See below” at the end of it.

Overall, this reviewer finds merit in the work performed by the authors. While the suggestions being made are non-trivial, this reviewer hopes the authors will find the comments beneficial to their developing, valuable and interesting story about H2-Ob as a gene switch in the AD pathway.

6. PLOS authors have the option to publish the peer review history of their article (what does this mean?). If published, this will include your full peer review and any attached files.

Reviewer #1: **Yes: **Nipun Chopra

---

## [Author Response · Author response to Decision Letter 0]

12 May 2023

Reviewer Comments:

The current work identifies a potential role for H2-Ob as a gene switch in the interaction between Csf1r and Milr1 gene pair. The work utilizes a rat model of AD, wherein stereotactic techniques were utilized to administer oligomerized A-beta into the hippocampus of rats. The sample sizes used (n=9 for both groups) were appropriate, and the authors utilize IHC, behavior and gene expression data to support their findings.

This review raises the following concerns about the manuscript and recommends the addressing of these concerns to the authors:

We would like to thank the reviewer for his valuable and constructive suggestions. In the following, we present a point-by-point response to each of the comments.

(Q1) Introduction: The introduction section is brief and would benefit from a literature review describing the five genes studied (namely, Csf1r, Milr1, H2-ob, Slamf6, Hexb).

(A1) We agree with the reviewer. In the revised manuscript, we described the role of the studied genes in AD (paragraph 2 on page 4). 

“On the role of H2-Ob, Csf1r, Milr1, Slc14a1, Slamf6 and Hexb in AD

The H2-Ob gene, also known as H2-O in humans, encodes a non-classical major histocompatibility complex (MHC) class II molecule that is expressed specifically by antigen-presenting cells such as macrophages, dendritic cells, and B cells [1]. Some evidence suggests that the H2-Ob gene may play a role in neurodegenerative diseases, including AD [2, 3]. It has been found that H2-Ob is up-regulated in Niemann-Pick disease type C and multiple sclerosis, two neurodegenerative disorders [4, 5]. Moreover, a recent study showed that the expression levels of MHC class II molecules were increased in a transgenic rat model of human tauopathy [6]. Furthermore, using a computational model, we previously suggested that the H2-Ob gene might regulate the co-expression relationship of two genes Csf1r and Milr1, possibly being involved in AD [7]. In addition, some experimental studies suggest that HLA-DOB is involved in AD [8]. These findings suggest that the H2-Ob gene may play a critical role in the pathogenesis of AD through its involvement in the immune response and microglial function. In addition, modulating the activation of non-neuronal cells that normally control neuronal sensitization, for example by targeting H2-Ob gene expression, opens new perspectives for the development of therapies that target neuroinflammation and AD progression [9]. Therefore, further research is needed to fully understand the mechanisms underlying the role of H2-Ob gene in the course of neurodegeneration.

The Csf1r gene, also known as colony-stimulating factor 1 receptor gene, plays a well-established role in the differentiation and survival of mononuclear phagocytes, including microglia in the CNS [10]. Several studies have shown that dysregulation of the Csf1r gene and its associated signaling pathway is linked to microglial dysfunction and, in turn, the pathogenesis of neurodegenerative diseases. More specifically, pharmacological inhibition of the Csf1r gene reduces overall neuroinflammation and prevents neuronal loss and memory impairment in the AD mice model [11, 12]. Another study showed that genetic variants of the Csf1r gene were associated with an increased risk of AD [13]. In addition, inhibition of CSF-1R has been proposed as a potential therapeutic strategy for neurodegenerative diseases, including AD [14-16].

The Mast Cell Immunoglobulin-Like Receptor 1 gene (MILR1) encodes the Allergin-1 protein, which is expressed on the surface of mast cells [17]. Allergin-1 contains an immunoreceptor tyrosine-based inhibitory motif (ITIM)-like domain that modulates mast cell activation via the FcεRI-mediated signaling pathway [18]. Although Allergin-1 is not reported to plays a role in the pathogenesis of neurodegenerative diseases, existing evidence suggests its potential indirect involvement. Amyloid peptides, as a hallmark of AD pathology, induce mast cell activation and degranulation response, resulting in the release of different pro-inflammatory substances [19, 20]. The release of such bioactive molecules may be associated with the onset (or even progression) of AD [21, 22]. On the other hand, Allergin-1 acts as a modulator protein for mast cell activation and also inhibits mast cell degranulation via suppression of FcεRI-mediated signaling [18]. Furthermore, the dysregulation of the FcεRI-mediated signaling pathway in AD pathology has been reported recently [7, 23]. Taken together, the role of the Milr1 gene (Allergin-1) in neurodegenerative diseases, especially AD, is sensible.

The Slamf6 gene, also known as CD352, is a member of the signaling lymphocyte activation molecule (SLAM) family of receptors. It is predominantly expressed on the surface of natural killer (NK) cells, T cells, and dendritic cells. The Slamf6 gene is involved in various physiological and pathological processes, regulating immune responses, NK-cell development and cytotoxicity, neutrophil functions, and trogocytosis [24]. Although the Slamf6 gene has been identified as a contributing factor to the progression of some autoimmune diseases [25-27], limited research has been done on its direct involvement in neurodegenerative diseases. However, a recent study suggested that dysregulation of in the Slamf6 gene may contribute to the development of multiple sclerosis [28].

The Hexb gene encodes the beta subunit of β-hexosaminidase, which is stably expressed in brain microglia. This lysosomal enzyme is involved in the breakdown of gangliosides. Mutations in the Hexb gene can result in the accumulation of glycosphingolipids, leading to various lysosomal storage disorders collectively known as GM2 gangliosidosis [29]. Lysosomal abnormalities are one of the hallmarks of AD, and they progress over time. According to a study, the accumulation of ganglioside-bound amyloid peptide, which is associated with AD, occurs due to lysosomal dysfunction in the β-hexosaminidase knockout mouse model [30]. Another study found that although heterozygosity of the Hexb gene reduced learning flexibility, such gene is haploinsufficient in the mouse AD brain [31]. Recent studies reveal that the Hexb gene is up-regulated in the AD brain at both transcriptome [32] and proteome levels [33].

The Slc14a1 gene, also known as the urea transporter UT-B, is responsible for the transport of urea across cell membranes in various organs, including the brain. The SLC14A1 gene is primarily expressed in astrocytes and ependymal cells, in which it is believed to play a central role in the maintenance of urea homeostasis of the nervous system [34]. Although the precise role of the Slc14a1 gene in neurodegenerative diseases remains poorly understood, several studies have reported the dysregulation of Slc14a1 expression in neurodegenerative diseases, including AD [7], HD [35] and ALS [36]. Moreover, disruption of urea metabolism and subsequent accumulation of urea in the brain has been reported in AD [37], whilst the Aβ-derived ammonia is reported to be detoxified by the urea cycle of astrocytes [38]. Furthermore, Slc14a1-deficient mice exhibit urea accumulation in the hippocampus, resulting in behavioral impairment, nitric oxide disruption and neuronal loss [39]. A recent study indicated that the expression level of the Slc14a1 gene regulates the inflammatory responses in microglial and neuroblastoma cells, following lipopolysaccharide exposure. Therefore, it has been suggested that the inhibition of the Slc14a1 gene can be a potential therapeutic target for AD [40].”

(Q2) Introduction: Similarly, a stronger background should be provided on the utility of the A-beta administration model in rats; while the authors confirm A-beta presence in the hippocampus, it is unclear whether this results in microglial and mast cell activation in these rats.

(A2) We agree with the reviewer that we should have explained this point. In the revised manuscript, we clarified it (paragraph 2 on page 7).

“Rodent models of AD

Amyloid-β (Aβ) is the primary pathological hallmark of familial and sporadic AD. The accumulation of Aβ oligomers in the brain can lead to the formation of plaques, which contribute to neuronal damage and cognitive decline. Clinical and experimental studies provide strong evidence that the acute increase in Aβ levels in the brain is responsible for inducing AD-like phenotypes [41, 42]. Therefore, animal models of AD have been developed based on neural overexpression or administration of Aβ. Two commonly used animal models to study AD are transgenic and non-transgenic rodent models. While transgenic rodent models are useful in mimicking human AD pathology, their development is time-consuming and expensive. Non-transgenic models, such as pathogen-induced AD models, can produce AD-like behavioral abnormalities, although may not accurately reflect the fundamental pathophysiology of AD. The Aβ-injected rodent model is a pathogen-induced AD model that exhibits Aβ pathology and is attractive for AD investigation due to its controllability. Researchers can control individual differences in the mice and enable timely drug treatment depending on the mechanism of the candidate drug [43]. Moreover, injecting Aβ peptide into the hippocampus has been associated with several characteristic features of AD, including inflammatory reactivity, neuronal loss, and vascular perturbations [44, 45]. Specifically, intra-hippocampal injection of Aβ peptide leads to inflammatory responses mediated by activated microglia, which is coupled with granule cell neuron loss in the rat AD model [46]. Finally, previous studies suggest that intracerebral injection of Aβ peptide can change the gene expression profile [47].”

(Q3) Results: The first three figures serve to establish the model (presence of a-beta and impact of a-beta on spatial learning), whereas the major claim of the paper are the subsequent figures. This reviewer recommends that the authors add additional figures associated with additional suggested work, and to add figures 1-3 (or some combination) to the supplementary data.

(A3) Thank you for your suggestion. We merged Figures (2 and 3) related to the Morris water maze test. We should have kept this figure as main Figure because we mentioned it in the discussion section.

Revised part of Figure legends

“Figure 2. Comparison of results of the Morris water maze test of the AD model rats and the control rats. (A) Comparison of the searching distance in the place navigation test; (B) Comparison of the swimming distance in the target quadrant in the probe test. (C) Comparisons of the escape latency (i.e., swimming time) in the place navigation test; (D) Comparison of the time spent in the target quadrant in the probe test. Data on each day are expressed as means ± standard error.”

(Q4) Results: The data in Figure 4b is not statistically significant, but the conclusion seems to suggest that it is. This should be clarified.

(A4) We agree with the reviewer. In the revised manuscript, we clarified this point in more details (paragraph 2 on Page 13). 

 “Interestingly, the expression level of the RT1-DOb gene in the AD hippocampus samples was higher than in control samples, and the difference approached statistical significance (p= 0.056) (Fig. 3B). Overall, the findings are consistent with the proposition that RT1-DOb might act as a switch gene in promoting AD.

It should be noted here that although the change in expression level of the Z gene in a Z/{X1, X2} triplet is a fundamental characteristic of such a switch gene, it may not be directly associated with the effectiveness of the switch gene. In other words, sometimes, a slight change in the expression level of a switch gene (Z) can lead to a drastic shift in the co-expression of the X1 and X2 gene pair.”

(Q5) Results: While the authors clarify that they do not have access to AD mice models (hence the rationale for using the rat model), have the authors confirmed this gene interaction in post-mortem human brain? This would substantially strengthen their argument, as for now, it seems far too limited to a highly unique non-genetic murine model.

(A5) This fact should be emphasized that in the previous study, we have confirmed the gene interactions involved in the two mentioned triplets in the mouse brain of a transgenic AD model. Also, we agree with the reviewer that verification of such interactions in the human brain would be a crucial step toward validating our findings. Nevertheless, the post-mortem AD brain is not useful for detecting the switching mechanisms which commence the disease. Indeed, to survey the switching mechanism, we need gene expression data corresponding to the early stage of AD. Furthermore, death is known to be an intervening factor per se in gene expression level [48, 49].

We explained in the above description in the “Conclusion and future work” section of revised manuscript (the end of paragraph 3, on page17).

Revised part of Conclusion and future work

“In addition, since the gene interactions involved in the two mentioned triplets have been defined in a mouse AD model [50], confirmation of those interactions in the human brain would be an importance step validating our findings. It should be noted here that switching mechanisms are associated with onset of disease. Therefore, the early stage of AD should be considered to validate these mechanisms.”

 (Q6) Results: There is an incongruity in the timeline. The behavioral model in Fig 2 and 3 shows that the presumable inhibitory learning effects of A-beta are lost by day 3 – both tests show no different b/w the model and control at day 3. While the authors do not specify this (they should), it appears the gene expression data is performed at day 20 (after the brains were obtained for IHC). Therefore, there needs to be an explanation for the long-term effects of A-beta on the gene switch absent of any behavioral differences.

(A6) We would like to thank the reviewer for bringing this issue to our attention. It should have been described in the manuscript. In the “Discussion” section of the revised manuscript, we explained it (paragraph 3, on page15).

 “The results obtained from the Morris water maze (MWM) test clearly demonstrate a significant difference in the searching distance (Fig. 2A) and the escape latency (Fig. 2C) between the control and AD model groups on the first and second days of the training phase. As shown in the figure, the control group reached the escape platform in less time and covered a shorter searching distance compared to the AD model group. This difference was more pronounced on the first day, less so on the second day, and even less so on the third day. The difference on the third day was not statistically significant. These results indicate that training phase has an improving effect on learning in AD model rats. This effect has been confirmed in other studies as well [51]. Based on these findings, it is suggested that keeping the brain active through mental activities such as reading or solving puzzles may slow down the progression of Alzheimer's disease. However, the results of this study confirm the difference between the control and Alzheimer's groups not only on the first and second days but also on the spatial probe test (Fig. 2B and 2D).”

(Q7) Results: While the authors discuss the role of gene switch, there is a complete absence of confirmation of protein data. The gene switch role presupposes the ability of H2-ob to act as a transcription factor protein. Therefore, at the very least, protein expression via western blotting is necessary to complement the RT-PCR RNA expression data. If the proteins do not have antibodies available, perhaps an indirect measurement (measuring the activity/levels of a downstream pathway) would suffice as supporting evidence.

(A7) We agree with the reviewer' point that the switch gene might act as a transcription factor protein. While we did not include protein data in our study, we acknowledge the importance of this analysis and plan to address it in future research. In the previous version of manuscript, we have mentioned this issue in the “Conclusion and future work” section (paragraph 2, on page17). 

“In the next step, the H2-Ob gene can be knocked down using a gene-knockdown technique such as RNA interference. Subsequently, the relationship between the H2-Ob gene and the {Csf1r, Milr1} gene pair be evaluated at the protein level using Western blotting.”

(Q8-1) Discussion: The discussion section is far too short and does not identify or speculate the mechanism of any of the work. The link between the gene switch and cellular response (microglia/mast) is minimal,

(A8-1) We agree with reviewer. We explained this issue in more details in the revised manuscript (paragraph 2 on Page 16).

“Studies that have directly investigated the role of the H2-Ob gene in microglia/mast cell response are rare, although some evidence suggests that it is important in microglial activation. The H2-Ob gene is a member of the major histocompatibility complex class II (MHC-II) antigens, which are expressed by antigen-presenting cells, including microglia[8]. MHC-II microglia plays a pivotal role in the activation of both innate and adaptive immune responses in neuroinflammation conditions, including AD [52-54]. Furthermore, dysregulation of H2-Ob may contribute to the pathogenesis of Parkinson's disease [55]. Although the above findings emphasize the role of MHC-II antigens in neuroinflammation pathogenesis, a conclusive association is yet to be found between H2-Ob expression and AD progression. Moreover, to the best of our knowledge, no experimental or computational evidence supports the claim that Aβ peptide regulates H2-Ob gene expression.”

(Q8-2) and an identification of the method by which A-beta affects gene expression is entirely absent. For example, the section would benefit from an explanation of a-beta affecting cellular processes – either directly (there is data that supports the ability of a-beta to act as a transcription factor) or indirectly (via activation of intracellular cascades) or non-canonical explanations (such as data that supports the ability of a-beta to inset itself directly into the plasma membrane of neurons).

(A8-2) We agree with reviewer. We explained this issue in the revised manuscript (paragraph 2 on Page 14).

“We used a rat Aβ model of Alzheimer's disease. The Aβ peptide, the primary component of senile plaques in AD, contributes to neurotoxicity and other pathogenic mechanisms in various ways. Firstly, it can act as a transcription factor and directly regulate genes involved in AD-related pathways. Some pathways that can be directly regulated by Aβ peptide include insulin signaling, actin cytoskeleton dynamics, steroid and lipid metabolism, neuronal cell death, and DNA binding [56-59]. In particular, the Aβ peptide influences the expression of Girk and Kcnq channel genes, which are related to the impairment of learning and memory in AD [60].

Secondly, the Aβ peptide exerts its toxic effect indirectly by triggering intracellular cascades, resulting in the impairment of synaptic function, neuronal signaling, mast cell and microglial activities, energy metabolism, and mitochondrial function. For instance, the toxicity levels of Aβ peptide lead to tau phosphorylation, and in turn, neuronal death via the PAX6 signaling pathway [61]. Moreover, this peptide affects microglial activities via changes in the calcium level, proinflammatory activity, and gliotransmission through the RAGE-NF-κB pathway [62]. Furthermore, the Aβ oligomers are reported to induce memory impairment in an acute mouse model of AD through Toll-like receptor 4-dependent glial cell activation [63]. Additionally, the Aβ peptide induces mast cell secretion via both a CD47/Pl-integrin membrane complex coupled with GJ-protein [20] and a pannexin1 hemichannel-dependent mechanism [19]. Also, the Aβ peptide activates downstream pathways related to mitochondrial damage in microglia through TREM2 (triggering receptor expressed in myeloid cells 2), which is coupled with aggravating inflammation and neurotoxicity [64].

Finally, there is strong evidence suggesting that Aβ peptide can play a pathogenic role in non-trivial ways. The Aβ peptide can independently induce uncontrolled, neurotoxic ion flux across cellular membranes, either by self-assembly to pores or by disrupting the lipid membrane. Specifically, exposure of the cell to Aβ leads to disrupting Ca2+ homeostasis and eventually elevated concentrations of intracellular calcium ions, which contribute to cognitive impairment and cell death [65, 66].”

(Q9-1) Discussion: Is there computational or bioinformatics evidence that A-beta may regulate RT1-Dob gene expression data? 

(A9-1) We agree with reviewer. We explained it in the revised part of Discussion (the end of paragraph 2 on Page 16).

“Moreover, to the best of our knowledge, no experimental or computational evidence supports the claim that Aβ peptide regulates H2-Ob gene expression.”

 (Q9-2) Similarly, is there such data that RT1-Dob could mechanistically regulate the gene pair?

(A9-2) We agree with reviewer. We explained it in the revised part of Introduction (paragraph 2 on Page 4).

“Furthermore, using a computational model, we previously suggested that the H2-Ob gene might regulate the co-expression relationship of two genes Csf1r and Milr1, possibly being involved in AD [7].”

(Q10) Proof-read: This is a minor concern, but the authors might benefit from proofreading the manuscript. For example, the author email is not provided, and the end of the first intro paragraph has a stand alone “See below” at the end of it.

(A10) Thank you for your hint. We read our manuscript again carefully, and found a number of typos and grammatical errors. We hope that the revised manuscript has been improved significantly. 

Overall, there is merit in the work performed by the authors. While the suggestions being made are non-trivial, it is anticipated that the authors will find the comments beneficial to their developing, valuable and interesting story about H2-Ob as a gene switch in the AD pathway.

Refrences:

1. Reith W, LeibundGut-Landmann S, Waldburger J-M. Regulation of MHC class II gene expression by the class II transactivator. Nature Reviews Immunology. 2005;5(10):793-806.

2. Orre M, Kamphuis W, Osborn LM, Jansen AH, Kooijman L, Bossers K, et al. Isolation of glia from Alzheimer's mice reveals inflammation and dysfunction. Neurobiology of aging. 2014;35(12):2746-60.

3. Kamphuis W, Kooijman L, Schetters S, Orre M, Hol EM. Transcriptional profiling of CD11c-positive microglia accumulating around amyloid plaques in a mouse model for Alzheimer's disease. Biochimica et Biophysica Acta (BBA)-Molecular Basis of Disease. 2016;1862(10):1847-60.

4. Liao G, Wen Z, Irizarry K, Huang Y, Mitsouras K, Darmani M, et al. Abnormal gene expression in cerebellum of Npc1-/- mice during postnatal development. Brain research. 2010;1325:128-40. Epub 2010/02/16. doi: 10.1016/j.brainres.2010.02.019. PubMed PMID: 20153740; PubMed Central PMCID: PMCPMC2848886.

5. Li H, Hou X, Liang Y, Xu F, Zhang X, Cui P, et al. Gene-based tests of a genome-wide association study dataset highlight novel multiple sclerosis risk genes. Frontiers in Neuroscience. 2021;15:614528.

6. Stozicka Z, Korenova M, Uhrinova I, Cubinkova V, Cente M, Kovacech B, et al. Environmental Enrichment Rescues Functional Deficit and Alters Neuroinflammation in a Transgenic Model of Tauopathy. Journal of Alzheimer's disease : JAD. 2020;74(3):951-64. Epub 2020/03/03. doi: 10.3233/jad-191112. PubMed PMID: 32116255.

7. Khayer N, Marashi S-A, Mirzaie M, Goshadrou F. Three-way interaction model to trace the mechanisms involved in Alzheimer’s disease transgenic mice. PloS one. 2017;12(9):e0184697.

8. Patel D, Zhang X, Farrell JJ, Chung J, Stein TD, Lunetta KL, et al. Cell-type-specific expression quantitative trait loci associated with Alzheimer disease in blood and brain tissue. Translational Psychiatry. 2021;11(1):250.

9. Skaper SD, Facci L, Giusti P. Mast cells, glia and neuroinflammation: partners in crime? Immunology. 2014;141(3):314-27. Epub 2013/09/17. doi: 10.1111/imm.12170. PubMed PMID: 24032675; PubMed Central PMCID: PMCPMC3930370.

10. Wei S, Nandi S, Chitu V, Yeung YG, Yu W, Huang M, et al. Functional overlap but differential expression of CSF‐1 and IL‐34 in their CSF‐1 receptor‐mediated regulation of myeloid cells. Journal of leukocyte biology. 2010;88(3):495-505.

11. Spangenberg EE, Lee RJ, Najafi AR, Rice RA, Elmore MR, Blurton-Jones M, et al. Eliminating microglia in Alzheimer's mice prevents neuronal loss without modulating amyloid-β pathology. Brain. 2016;139(Pt 4):1265-81. Epub 2016/02/28. doi: 10.1093/brain/aww016. PubMed PMID: 26921617; PubMed Central PMCID: PMCPMC5006229.

12. Dagher NN, Najafi AR, Kayala KMN, Elmore MR, White TE, Medeiros R, et al. Colony-stimulating factor 1 receptor inhibition prevents microglial plaque association and improves cognition in 3xTg-AD mice. Journal of neuroinflammation. 2015;12:1-14.

13. Giau VV, Senanarong V, Bagyinszky E, An SSA, Kim S. Analysis of 50 neurodegenerative genes in clinically diagnosed early-onset Alzheimer’s disease. International journal of molecular sciences. 2019;20(6):1514.

14. Han J, Chitu V, Stanley ER, Wszolek ZK, Karrenbauer VD, Harris RA. Inhibition of colony stimulating factor-1 receptor (CSF-1R) as a potential therapeutic strategy for neurodegenerative diseases: opportunities and challenges. Cellular and Molecular Life Sciences. 2022;79(4):219.

15. Neal ML, Fleming SM, Budge KM, Boyle AM, Kim C, Alam G, et al. Pharmacological inhibition of CSF1R by GW2580 reduces microglial proliferation and is protective against neuroinflammation and dopaminergic neurodegeneration. FASEB journal: official publication of the Federation of American Societies for Experimental Biology. 2020;34(1):1679.

16. Olmos-Alonso A, Schetters ST, Sri S, Askew K, Mancuso R, Vargas-Caballero M, et al. Pharmacological targeting of CSF1R inhibits microglial proliferation and prevents the progression of Alzheimer’s-like pathology. Brain. 2016;139(3):891-907.

17. Nagai K, Tahara-Hanaoka S, Morishima Y, Tokunaga T, Imoto Y, Noguchi E, et al. Expression and function of Allergin-1 on human primary mast cells. PLoS One. 2013;8(10):e76160.

18. Hitomi K, Tahara-Hanaoka S, Someya S, Fujiki A, Tada H, Sugiyama T, et al. An immunoglobulin-like receptor, Allergin-1, inhibits immunoglobulin E-mediated immediate hypersensitivity reactions. Nat Immunol. 2010;11(7):601-7. Epub 2010/06/08. doi: 10.1038/ni.1886. PubMed PMID: 20526344.

19. Harcha PA, Vargas A, Yi C, Koulakoff AA, Giaume C, Sáez JC. Hemichannels are required for amyloid β-peptide-induced degranulation and are activated in brain mast cells of APPswe/PS1dE9 mice. Journal of Neuroscience. 2015;35(25):9526-38.

20. Niederhoffer N, Levy R, Sick E, Andre P, Coupin G, Lombard Y, et al. Amyloid β peptides trigger CD47-dependent mast cell secretory and phagocytic responses. International Journal of Immunopathology and Pharmacology. 2009;22(2):473-83.

21. Harcha PA, Garcés P, Arredondo C, Fernández G, Sáez JC, van Zundert B. Mast cell and astrocyte hemichannels and their role in Alzheimer’s disease, ALS, and harmful stress conditions. International Journal of Molecular Sciences. 2021;22(4):1924.

22. Folch J, Petrov D, Ettcheto M, Pedros I, Abad S, Beas-Zarate C, et al. Masitinib for the treatment of mild to moderate Alzheimer’s disease. Expert review of neurotherapeutics. 2015;15(6):587-96.

23. Chen Y, Liu Q, Liu J, Wei P, Li B, Wang N, et al. Revealing the modular similarities and differences among Alzheimer’s disease, vascular dementia, and Parkinson’s disease in genomic networks. NeuroMolecular Medicine. 2021:1-14.

24. Yigit B, Wang N, Herzog RW, Terhorst C. SLAMF6 in health and disease: Implications for therapeutic targeting. Clinical immunology (Orlando, Fla). 2019;204:3-13. Epub 2018/10/27. doi: 10.1016/j.clim.2018.10.013. PubMed PMID: 30366106; PubMed Central PMCID: PMCPMC8969216.

25. Humbel M, Bellanger F, Horisberger A, Suffiotti M, Fluder N, Makhmutova M, et al. SLAMF Receptor Expression Identifies an Immune Signature That Characterizes Systemic Lupus Erythematosus. Frontiers in immunology. 2022;13:843059. Epub 2022/05/24. doi: 10.3389/fimmu.2022.843059. PubMed PMID: 35603218; PubMed Central PMCID: PMCPMC9120573.

26. Savola P, Kelkka T, Rajala HL, Kuuliala A, Kuuliala K, Eldfors S, et al. Somatic mutations in clonally expanded cytotoxic T lymphocytes in patients with newly diagnosed rheumatoid arthritis. Nature communications. 2017;8:15869. Epub 2017/06/22. doi: 10.1038/ncomms15869. PubMed PMID: 28635960; PubMed Central PMCID: PMCPMC5482061 Bristol-Myers Squibb and Pfizer. S.M. has received research funding from Ariad. J.S. has received lecture fees from Roche. The remaining authors declare no competing financial interests.

27. Wang N, Keszei M, Halibozek P, Yigit B, Engel P, Terhorst C. Slamf6 negatively regulates autoimmunity. Clinical immunology (Orlando, Fla). 2016;173:19-26. Epub 2016/11/05. doi: 10.1016/j.clim.2016.06.009. PubMed PMID: 27368806; PubMed Central PMCID: PMCPMC5206809.

28. Schnell A, Huang L, Singer M, Singaraju A, Barilla RM, Regan BML, et al. Stem-like intestinal Th17 cells give rise to pathogenic effector T cells during autoimmunity. Cell. 2021;184(26):6281-98 e23. Epub 2021/12/08. doi: 10.1016/j.cell.2021.11.018. PubMed PMID: 34875227; PubMed Central PMCID: PMCPMC8900676.

29. Masuda T, Amann L, Sankowski R, Staszewski O, Lenz M, d´ Errico P, et al. Novel Hexb-based tools for studying microglia in the CNS. Nature immunology. 2020;21(7):802-15.

30. Keilani S, Lun Y, Stevens AC, Williams HN, Sjoberg ER, Khanna R, et al. Lysosomal dysfunction in a mouse model of Sandhoff disease leads to accumulation of ganglioside-bound amyloid-β peptide. The Journal of neuroscience : the official journal of the Society for Neuroscience. 2012;32(15):5223-36. Epub 2012/04/13. doi: 10.1523/jneurosci.4860-11.2012. PubMed PMID: 22496568; PubMed Central PMCID: PMCPMC6622109.

31. Whyte LS, Fourrier C, Hassiotis S, Lau AA, Trim PJ, Hein LK, et al. Lysosomal gene Hexb displays haploinsufficiency in a knock-in mouse model of Alzheimer's disease. IBRO neuroscience reports. 2022;12:131-41. Epub 2022/02/12. doi: 10.1016/j.ibneur.2022.01.004. PubMed PMID: 35146484; PubMed Central PMCID: PMCPMC8819126.

32. Sierksma A, Lu A, Mancuso R, Fattorelli N, Thrupp N, Salta E, et al. Novel Alzheimer risk genes determine the microglia response to amyloid-β but not to TAU pathology. EMBO molecular medicine. 2020;12(3):e10606. Epub 2020/01/18. doi: 10.15252/emmm.201910606. PubMed PMID: 31951107; PubMed Central PMCID: PMCPMC7059012.

33. Rangaraju S, Dammer EB, Raza SA, Gao T, Xiao H, Betarbet R, et al. Quantitative proteomics of acutely-isolated mouse microglia identifies novel immune Alzheimer's disease-related proteins. Molecular neurodegeneration. 2018;13(1):34. Epub 2018/06/30. doi: 10.1186/s13024-018-0266-4. PubMed PMID: 29954413; PubMed Central PMCID: PMCPMC6025801.

34. Recabarren D, Alarcón M. Gene networks in neurodegenerative disorders. Life sciences. 2017;183:83-97. Epub 2017/06/18. doi: 10.1016/j.lfs.2017.06.009. PubMed PMID: 28623007.

35. Handley RR, Reid SJ, Brauning R, Maclean P, Mears ER, Fourie I, et al. Brain urea increase is an early Huntington's disease pathogenic event observed in a prodromal transgenic sheep model and HD cases. Proceedings of the National Academy of Sciences of the United States of America. 2017;114(52):E11293-e302. Epub 2017/12/13. doi: 10.1073/pnas.1711243115. PubMed PMID: 29229845; PubMed Central PMCID: PMCPMC5748180.

36. Recabarren-Leiva D, Alarcón M. New insights into the gene expression associated to amyotrophic lateral sclerosis. Life sciences. 2018;193:110-23. Epub 2017/12/16. doi: 10.1016/j.lfs.2017.12.016. PubMed PMID: 29241710.

37. Hansmannel F, Sillaire A, Kamboh MI, Lendon C, Pasquier F, Hannequin D, et al. Is the urea cycle involved in Alzheimer's disease? Journal of Alzheimer's disease : JAD. 2010;21(3):1013-21. Epub 2010/08/10. doi: 10.3233/jad-2010-100630. PubMed PMID: 20693631; PubMed Central PMCID: PMCPMC2945690.

38. Ju YH, Bhalla M, Hyeon SJ, Oh JE, Yoo S, Chae U, et al. Astrocytic urea cycle detoxifies Aβ-derived ammonia while impairing memory in Alzheimer’s disease. Cell Metabolism. 2022;34(8):1104-20. e8.

39. Li X, Ran J, Zhou H, Lei T, Zhou L, Han J, et al. Mice lacking urea transporter UT-B display depression-like behavior. Journal of molecular neuroscience : MN. 2012;46(2):362-72. Epub 2011/07/14. doi: 10.1007/s12031-011-9594-3. PubMed PMID: 21750947.

40. Jones AC, Pinki F, Stewart GS, Costello DA. Inhibition of Urea Transporter (UT)-B Modulates LPS-Induced Inflammatory Responses in BV2 Microglia and N2a Neuroblastoma Cells. Neurochemical research. 2021;46(6):1322-9. Epub 2021/03/07. doi: 10.1007/s11064-021-03283-4. PubMed PMID: 33675462.

41. Bloom GS. Amyloid-β and tau: the trigger and bullet in Alzheimer disease pathogenesis. JAMA neurology. 2014;71(4):505-8. Epub 2014/02/05. doi: 10.1001/jamaneurol.2013.5847. PubMed PMID: 24493463.

42. Gouras GK, Olsson TT, Hansson O. β-Amyloid peptides and amyloid plaques in Alzheimer's disease. Neurotherapeutics : the journal of the American Society for Experimental NeuroTherapeutics. 2015;12(1):3-11. Epub 2014/11/06. doi: 10.1007/s13311-014-0313-y. PubMed PMID: 25371168; PubMed Central PMCID: PMCPMC4322079.

43. Kim HY, Lee DK, Chung B-R, Kim HV, Kim Y. Intracerebroventricular injection of amyloid-β peptides in normal mice to acutely induce Alzheimer-like cognitive deficits. JoVE (Journal of Visualized Experiments). 2016;(109):e53308.

44. Jantaratnotai N, Ryu JK, Schwab C, McGeer PL, McLarnon JG. Comparison of vascular perturbations in an Aβ-injected animal model and in AD brain. International Journal of Alzheimer’s Disease. 2011;2011.

45. McGeer EG, McGeer PL. The importance of inflammatory mechanisms in Alzheimer disease. Experimental gerontology. 1998;33(5):371-8.

46. McLarnon JG. Correlated inflammatory responses and neurodegeneration in peptide-injected animal models of Alzheimer’s disease. BioMed research international. 2014;2014.

47. Kong L-n, Zuo P-p, Mu L, Liu Y-y, Yang N. Gene expression profile of amyloid beta protein-injected mouse model for Alzheimer disease. Acta Pharmacologica Sinica. 2005;26(6):666-72.

48. Tomita H, Vawter MP, Walsh DM, Evans SJ, Choudary PV, Li J, et al. Effect of agonal and postmortem factors on gene expression profile: quality control in microarray analyses of postmortem human brain. Biological psychiatry. 2004;55(4):346-52.

49. Franz H, Ullmann C, Becker A, Ryan M, Bahn S, Arendt T, et al. Systematic analysis of gene expression in human brains before and after death. Genome biology. 2005;6(13):1-9.

50. Khayer N, Marashi SA, Mirzaie M, Goshadrou F. Three-way interaction model to trace the mechanisms involved in Alzheimer's disease transgenic mice. PLoS One. 2017;12(9):e0184697. Epub 2017/09/22. doi: 10.1371/journal.pone.0184697. PubMed PMID: 28934252.

51. Soheili Kashani M, Salami M, Rezaei-Tavirani M, Talaei Zavareh SA. Maze training improves learning in an Alzheimer model of rat. KAUMS Journal (FEYZ). 2010;14(3):209-16.

52. Rocha NP, De Miranda AS, Teixeira AL. Insights into neuroinflammation in Parkinson’s disease: from biomarkers to anti-inflammatory based therapies. BioMed research international. 2015;2015.

53. Boyko AA, Troyanova NI, Kovalenko EI, Sapozhnikov AM. Similarity and differences in inflammation-related characteristics of the peripheral immune system of patients with Parkinson’s and Alzheimer’s diseases. International journal of molecular sciences. 2017;18(12):2633.

54. Chen Y, Colonna M. Microglia in Alzheimer's disease at single-cell level. Are there common patterns in humans and mice? The Journal of experimental medicine. 2021;218(9). Epub 2021/07/23. doi: 10.1084/jem.20202717. PubMed PMID: 34292312; PubMed Central PMCID: PMCPMC8302448 study and received research support from Alector, Amgen, Ono, and Pfizer; in addition, M. Colonna is a scientific advisory board member of Vigil and NGMBio, is a consultant for Cell Signaling Technologies, and has a patent to TREM2 pending. No other disclosures were reported.

55. Hirsch EC, Vyas S, Hunot S. Neuroinflammation in Parkinson's disease. Parkinsonism & related disorders. 2012;18:S210-S2.

56. Malik B, Fernandes C, Killick R, Wroe R, Usardi A, Williamson R, et al. Oligomeric amyloid-β peptide affects the expression of genes involved in steroid and lipid metabolism in primary neurons. Neurochemistry international. 2012;61(3):321-33.

57. Martínez T, Pascual A. Gene expression profile in β-amyloid-treated SH-SY5Y neuroblastoma cells. Brain research bulletin. 2007;72(4-6):225-31.

58. Sebollela A, Freitas-Correa L, Oliveira FF, Paula-Lima AC, Saraiva LM, Martins SM, et al. Amyloid-β oligomers induce differential gene expression in adult human brain slices. Journal of Biological Chemistry. 2012;287(10):7436-45.

59. Barucker C, Sommer A, Beckmann G, Eravci M, Harmeier A, Schipke CG, et al. Alzheimer amyloid peptide Aβ 42 regulates gene expression of transcription and growth factors. Journal of Alzheimer's Disease. 2015;44(2):613-24.

60. Mayordomo-Cava J, Yajeya J, Navarro-Lopez JD, Jimenez-Diaz L. Amyloid-β (25-35) modulates the expression of GirK and KCNQ channel genes in the hippocampus. PLoS One. 2015;10(7):e0134385.

61. Zhang Y, Zhang Y, Aman Y, Ng CT, Chau W-H, Zhang Z, et al. Amyloid-β toxicity modulates tau phosphorylation through the PAX6 signalling pathway. Brain. 2021;144(9):2759-70.

62. Rodríguez-Giraldo M, González-Reyes RE, Ramírez-Guerrero S, Bonilla-Trilleras CE, Guardo-Maya S, Nava-Mesa MO. Astrocytes as a Therapeutic Target in Alzheimer’s Disease–Comprehensive Review and Recent Developments. International Journal of Molecular Sciences. 2022;23(21):13630.

63. Calvo-Rodriguez M, García-Rodríguez C, Villalobos C, Núñez L. Role of toll like receptor 4 in Alzheimer’s disease. Frontiers in immunology. 2020;11:1588.

64. Agrawal I, Jha S. Mitochondrial dysfunction and Alzheimer’s disease: Role of microglia. Frontiers in aging neuroscience. 2020:252.

65. Capone R, Quiroz FG, Prangkio P, Saluja I, Sauer AM, Bautista MR, et al. Amyloid-β-induced ion flux in artificial lipid bilayers and neuronal cells: resolving a controversy. Neurotoxicity research. 2009;16:1-13.

66. Shirwany NA, Payette D, Xie J, Guo Q. The amyloid beta ion channel hypothesis of Alzheimer’s disease. Neuropsychiatric disease and treatment. 2007;3(5):597-612.

---

## [Decision Letter · Decision Letter 1]

20 Jun 2023

RT-DOb , a switch gene for the gene pair {Csf1r, Milr1}, can influence the onset of Alzheimer’s disease by regulating communication between mast cell and microglia

PONE-D-22-24268R1

Dear Dr. Khayer,

We’re pleased to inform you that your manuscript has been judged scientifically suitable for publication and will be formally accepted for publication once it meets all outstanding technical requirements.

Kind regards,

Chandler L. Walker

Academic Editor

PLOS ONE

Additional Editor Comments (optional):

The authors have satisfactorily addressed prior reviewer's comments and concerns. The manuscript is now acceptable for publication.

Reviewers' comments:

Reviewer's Responses to Questions

**Comments to the Author**

1. If the authors have adequately addressed your comments raised in a previous round of review and you feel that this manuscript is now acceptable for publication, you may indicate that here to bypass the “Comments to the Author” section, enter your conflict of interest statement in the “Confidential to Editor” section, and submit your "Accept" recommendation.

Reviewer #1: All comments have been addressed

2. Is the manuscript technically sound, and do the data support the conclusions?

Reviewer #1: Yes

3. Has the statistical analysis been performed appropriately and rigorously? 

Reviewer #1: Yes

4. Have the authors made all data underlying the findings in their manuscript fully available?

Reviewer #1: Yes

5. Is the manuscript presented in an intelligible fashion and written in standard English?

Reviewer #1: Yes

6. Review Comments to the Author

Reviewer #1: Thank you to the authors for their revision. I find the manuscript significantly strenghtened and recommend it for approval. One small (but important) concern - please utilize high-definition images of your data. The current versions are blurry.

7. PLOS authors have the option to publish the peer review history of their article (what does this mean?). If published, this will include your full peer review and any attached files.

Reviewer #1: **Yes: **Nipun Chopra

---

## [Editor Report · Acceptance letter]

27 Jun 2023

PONE-D-22-24268R1 

<i>RT-DOb<i>, a switch gene for the gene pair {*Csf1r, Milr1*}, can influence the onset of Alzheimer’s disease by regulating communication between mast cell and microglia 

Dear Dr. Khayer:

I'm pleased to inform you that your manuscript has been deemed suitable for publication in PLOS ONE. Congratulations! Your manuscript is now with our production department. 

Kind regards, 

on behalf of

Dr. Chandler L. Walker 

Academic Editor

PLOS ONE